# Controlled multi-photon subtraction with cascaded Rydberg superatoms as single-photon absorbers

Nina Stiesdal [1], Hannes Busche[1], Kevin Kleinbeck [2], Jan Kumlin [2], Mikkel G. Hansen[1], Hans Peter Büchler [2] & Sebastian Hofferberth [1,3 ✉]

The preparation of light pulses with well-defined quantum properties requires precise control at the individual photon level. Here, we demonstrate exact and controlled multi-photon subtraction from incoming light pulses. We employ a cascaded system of tightly confined cold atom ensembles with strong, collectively enhanced coupling of photons to Rydberg states. The excitation blockade resulting from interactions between Rydberg atoms limits photon absorption to one per ensemble and rapid dephasing of the collective excitation suppresses stimulated re-emission of the photon. We experimentally demonstrate subtraction with up to three absorbers. Furthermore, we present a thorough theoretical analysis of our scheme where we identify weak Raman decay of the long-lived Rydberg state as the main source of infidelity in the subtracted photon number and investigate the performance of the multi-photon subtractor for increasing absorber numbers in the presence of Raman decay.

[1] Department of Physics, Chemistry and Pharmacy, Physics@SDU, University of Southern Denmark, Odense, Denmark. [2] Institute for Theoretical Physics III and Center for Integrated Quantum Science and Technology, University of Stuttgart, Stuttgart, Germany. [3] Institut für Angewandte Physik, University of Bonn, Bonn, Germany. ✉email: hofferberth@sdu.dk

Future optical quantum technology relies on precise control over the quantum state of light. Deterministic removal of exactly one, or more generally exactly $n_{sub}$, photons enables applications such as state-preparation for optical quantum simulation and computing[1–5] or quantum-enhanced metrology[6]. Photon subtraction can also give insight into more fundamental aspects of quantum optics[7]. Heralded single-photon subtraction[8] can be implemented using highly imbalanced beamsplitters[7,9], but the probabilistic nature limits the scalability of this approach[3,9,10]. Individual absorbers like a single two-level atom in free space seem well-suited for photon subtraction, as they are saturated by just one photon, but this approach is limited by weak atom–photon coupling, stimulated emission and short lifetimes of the saturated state. These problems can be mitigated by enhancing the atom–light coupling using a resonator, and transfer of the absorber to a third, dark state[11–13] not coupled to the incoming light as demonstrated with single atoms coupled to a microsphere resonator[14].

Strong photon–emitter coupling can also be achieved without optical resonators in atomic ensembles with collectively excited and long-lived Rydberg states, also referred to as Rydberg superatoms[15]. Rydberg atoms interact strongly with each other[16] and the resulting excitation blockade[17] can be mapped onto light fields to create strong optical nonlinearities at the single-photon level[18–22]. This has enabled many technical applications such as single-photon sources[23,24], optical transistors[25,26], removal of photons from stored light pulses[27,28], and photon–photon quantum gates[29] with recent efforts to combine these into multi-node networks[30–32]. Photon subtraction can also be realised using Rydberg superatoms as saturable single-photon absorbers[33,34] combining the blockade, which prevents multi-photon absorption, with rapid dephasing of the superatom into dark collective states to avoid stimulated re-emission.

In this work, we demonstrate a cascaded quantum system of up to three Rydberg superatom absorbers for controlled subtraction of specific photon numbers from a light pulse. In addition to demonstrating controlled multi-photon subtraction, we find that Raman decay is the main source of deviations from the ideal absorber behaviour. This is supported by a detailed theoretical analysis, which also shows that scaling beyond $n_{sub} = 3$ absorbers with high probability to subtract exactly $n_{sub}$ photons is realistic as long as Raman decay can be suppressed and a sufficiently high single-photon coupling is maintained.

## Results

**Implementation**. Figure 1a illustrates the implementation of individual Rydberg superatoms as a saturable single-photon absorber[33,34]. Weak pulses of probe light at $\lambda_p \approx 780$ nm, from which photons are to be subtracted, propagate through a small, optically thick ensemble of optically trapped $^{87}$Rb atoms. In combination with a strong, co-propagating control field at $\lambda_c \approx 480$ nm, the probe light couples the atomic ground state $|g\rangle = |5S_{1/2}, F = 2, m_F = 2\rangle$ to a Rydberg state $|r\rangle = |121S_{1/2}, m_J = 1/2\rangle$ via $|e\rangle = |5P_{3/2}, F = 3, m_F = 3\rangle$ in a Raman scheme (Fig. 1b). As a result of the Raman detuning $\Delta/2\pi \approx 100$ MHz, probe photons are only absorbed by the ensemble if the control field is tuned onto Raman resonance with two-photon detuning $\delta = 0$. Strong van der Waals interactions between Rydberg atoms lead to the blockade effect that suppresses multiple Rydberg excitations for atoms separated by $r < r_B$, where $r_B$ is the blockade radius that characterises the volume inside which the energy shift $V = C_6/r^6$ defined by the van der Waals coefficient $C_6$ exceeds the excitation linewidth.

If the radial probe beam waist ($1/e^2$-waist radius $\approx 6.5$ μm) and the extent of the ensemble along the probe axis constrain the

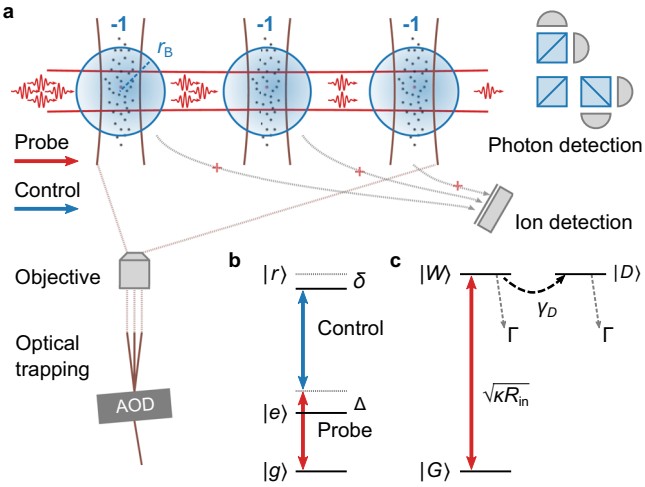

**Fig. 1 Realisation of up to three cascaded single-photon absorbers using Rydberg superatoms. a** To create $n_{sub}$ saturable superatom absorbers, we place $n_{sub}$ ensembles of cold $^{87}$Rb atoms in the path of a tightly focussed probe beam. Using an acousto-optical deflector (AOD), we can control the number and position of the optical traps that tightly confine the ensembles below the Rydberg blockade radius $r_B$ along the probe direction. **b** Within $r_B$ strong van der Waals interactions restrict each ensemble to a single Rydberg excitation as the probe photons and a control field couple $|g\rangle = |5S_{1/2}, F = 2, m_F = 2\rangle$ to a Rydberg state $|r\rangle = |121S_{1/2}, m_J = 1/2\rangle$ via $|e\rangle = |5P_{3/2}, F = 3, m_F = 3\rangle$ in a Raman scheme with detuning $\Delta/2\pi \approx 100$ MHz and thus to the absorption of a single photon at a time for a two-photon detuning of $\delta = 0$. The transmitted probe pulses are coupled into a single-mode optical fibre (not shown) and detected on four single-photon counters in a Hanbury-Brown-Twiss configuration. **c** Representation of the absorber as an effective three-level system in terms of singly excited collective states following adiabatic elimination of $|e\rangle$. Strong dephasing $\gamma_D$ from the bright excited state $|W\rangle$, with strong coupling $\sqrt{\kappa R_{in}}$ from the ground state $|G\rangle$, into dark excited states $|D\rangle$ prevents stimulated re-emission of the absorbed photon and the absorption of further photons until it is subject to Raman decay $\Gamma \ll \gamma_D, \kappa$.

excitation volume to a single Rydberg excitation, the superatom is saturated after absorbing just one photon. Consequently, all $N$ atoms in the excitation volume share the excitation in a collective bright state $|W\rangle = \sum_j |g_1 g_2 ... r_j ... g_N\rangle/\sqrt{N}$ with strongly enhanced collective coupling $\sqrt{\kappa R_{in}}$ from the collective ground state $|G\rangle = |g_1 g_2 ... g_j ... g_N\rangle$ where $\kappa = \sqrt{N} g_0 \Omega_c/(2\Delta)$[15,35–38]. Here, $g_0 \Omega_c/(2\Delta)$ is the effective single-probe-photon–single-atom coupling strength between $|g\rangle$ and $|r\rangle$, where $g_0$ is the single-probe-photon–single-atom coupling strength between $|g\rangle$ and $|e\rangle$, $R_{in}$ the incoming probe photon rate and $\Omega_c$ the control Rabi frequency. Following absorption of a photon, $|W\rangle$ dephases with rate $\gamma_D$ into a manifold of $N - 1$ collective dark states $\{|D\rangle\}$ that are orthogonal to $|W\rangle$ and no longer couple to the probe such that stimulated emission is suppressed, while maintaining the blockade. Besides dephasing, the excited collective states are also subject to decay of $|r\rangle$ with Raman decay $\Gamma = \Gamma_e \Omega_c^2/(2\Delta)^2$ as the dominant contribution, with $\Gamma_e$ being the natural linewidth of $|e\rangle$. Following adiabatic elimination of $|e\rangle$, the superatom dynamics can be described in terms of just $|W\rangle$, $|G\rangle$ and $|D\rangle$, a single dark state into which we condense all collective states in $\{|D\rangle\}$[15,39]. This effective three-level system will form the foundation of our theoretical analysis.

To implement multi-photon subtraction, we place $n_{sub}$ ensembles along the path of the probe and control fields (Fig. 1a)

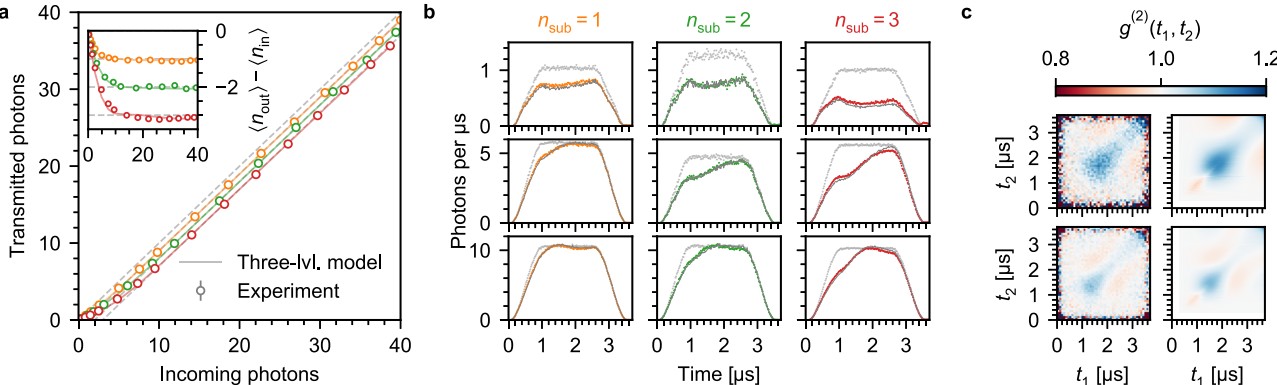

**Fig. 2 Effect of one, two and three single-photon absorbers on the probe transmission. a** Mean transmitted photons vs. mean incoming photon number. The inset shows the mean subtracted photons for the same data. All data are corrected for off-resonant absorption of the probe light and background counts. Error bars correspond to one standard error and are smaller than the markers for most data points. **b** Temporal profiles of probe light following transmission through $n_{sub} = 1$ (left), 2 (centre) and 3 (right) absorbers for different mean photon numbers. The incoming pulses recorded in the absence of the superatom absorbers are shown in light grey. Besides the experimental data, we also show the results of the three-level model fitted to the experimental data (dark grey). **c**. Photon correlations following transmission through three superatom absorbers. Second-order correlation functions $g^{(2)}(t_1, t_2)$ for $R_{in} \approx 5.5$ (top) and 10 $\mu s^{-1}$ (bottom). Besides the experimental data (left), we also show the predictions of the three-level model (right).

at distances $> 50\,\mu m \gg r_B$ such that the superatoms do not blockade each other and act independently. We trap the ensembles at the foci of individual optical trapping beams that intersect perpendicularly with the probe and confine each ensemble within $r_B$ along the probe axis. We use an acousto-optical deflector (AOD) to control the position and number of ensembles via the number and frequencies of radio-frequency (RF) signals applied[40–42]. A reservoir dipole trap (not shown in Fig. 1a) provides additional radial confinement ('Methods'). The ensembles have temperatures of $\approx 10\,\mu K$ with an extent $< r_B$ along the probe axis and $N \sim 10^4$ with exact numbers varying with $n_{sub} = 3, 2, 1$, respectively, due to variations in the trapping and cooling dynamics during ensemble preparation.

**Experimental results**. In the following, we experimentally demonstrate controlled subtraction of up to three photons by placing the corresponding number of absorbers $n_{sub}$ in the probe path. We measure the transmission for coherent, Tukey-shaped probe pulses with a pulse length of $\tau = 2.5\,\mu s$ (FWHM, with 1.0 $\mu s$ rise/fall time) and $\langle n_{in} \rangle \leq 40$, where $\langle n_{in} \rangle$ is the mean incoming photon number per pulse, using four single-photon counters in a Hanbury-Brown-Twiss configuration located behind a single-mode fibre. This configuration allows to analyse second-order photon correlations by calculating the correlation between all distinct pairs and averaging over the results. Without control field, we measure combined optical depths of the ensembles of $\approx 11$, 16 and 20 for $n_{sub} = 1$, 2 and 3 respectively, and find a probe transmission of $> 0.99$ which is slightly reduced at finite $\Delta/2\pi \approx$ 100 MHz due to off-resonant scattering of the probe light and which the data below are corrected for.

First, we investigate the difference between $\langle n_{in} \rangle$ and the mean transmitted photon number $\langle n_{out} \rangle$ (Fig. 2a). For $\langle n_{in} \rangle > 10$, we observe the expected reduction by $n_{sub}$, while we subtract fewer photons for $\langle n_{in} \rangle < 10$. This behaviour is expected, as for low $R_{in} \propto \langle n_{in} \rangle$, the pulse area $\sqrt{\kappa R_{in}} \tau$ is insufficient to drive the superatom population predominantly into $|W\rangle$ and $|D\rangle$. This becomes particularly evident in the shape of the transmitted pulses for $R_{in} \approx 1\,\mu s^{-1}$ (top row in Fig. 2b) with transmission well below one at their end, whereas we observe the onset of saturation for $R_{in} \approx 5\,\mu s^{-1}$ (centre row). Importantly, the duration to reach saturation increases with $n_{sub}$, because the driving between $|G\rangle$ and $|W\rangle$ reduces alongside the probe intensity following each

absorber. For $R_{in} \approx 10\,\mu s^{-1}$ (bottom row), saturation sets in even faster, but we observe a slight oscillation in the subsequent transmission, which reflects the superatom dynamics as the probe drives Rabi oscillations between $|G\rangle$ and $|W\rangle$ with strong damping due to $\gamma_D$[15]. To suppress superradiant re-emission of absorbed photons in the forward direction after the probe pulse[43,44], $\gamma_D$ has to be sufficiently strong not only compared to $1/\tau$, but also compared to the coherent dynamics[15,33]. The dephasing is dominated by atomic motion, which is enhanced by the co-propagating probe and control beams compared to a counter-propagating configuration, with additional contributions from elastic scattering of the Rydberg electron by ground-state atoms[45–47] and the non-uniform AC-Stark shift induced by the trapping light ('Methods'), which can only be compensated for on average. To characterise the system, we determine $\kappa$, $\gamma_D$ and $\Gamma$ by comparing the observed transmission to the predictions of a model of $n_{sub}$ effective three-level atoms strongly coupled to a chiral waveguide ('Methods'), assuming that $\kappa$, $\gamma_D$ and $\Gamma$ are equal for all absorbers. The results of the model are in good agreement with the experiment for both the subtracted photons (Fig. 2a) and pulse shape of the transmitted light (Fig. 2b) for $\{\kappa, \Gamma, \gamma_D\} = \{0.49, 0.045, 2.3\}\,\mu s^{-1}$ for $n_{sub} = 1$, $\{0.33, 0.020, 3.2\}\,\mu s^{-1}$ for $n_{sub} = 2$, and $\{0.35, 0.040, 2.4\}\,\mu s^{-1}$ for $n_{sub} = 3$.

A closer look at the number of subtracted photons (inset in Fig. 2a) reveals that it slightly exceeds $n_{sub}$ at high $\langle n_{in} \rangle$ indicating that a single absorber may subtract multiple photons. This excess cannot be explained by the deexcitation of the absorbers in the Rabi oscillation cycle as the associated spontaneous emission occurs back into the probe mode in forward direction with rate $\kappa$. Instead, it can be attributed to the small, but non-zero Raman decay $0 < \Gamma < \kappa$, $\gamma_D$, which leads to spontaneous re-emission in random, rather than the forward direction. Its presence, even if weak, however, leads to reduced fidelity to subtract exactly one photon per absorber as we will discuss in detail in our theoretical analysis.

To further demonstrate manipulation of the quantum state of light at the single-photon level, we also investigate the effect of the subtraction on the correlations between transmitted photons. Figure 2c shows the second-order correlation function $g^{(2)}(t_1, t_2) = \langle n(t_1)n(t_2)\rangle/(\langle n(t_1)\rangle\langle n(t_2)\rangle)$ for two of the transmitted pulses shown in Fig. 2b and $n_{sub} = 3$ alongside the theory prediction for three cascaded three-level absorbers. Following initial anti-bunching, we observe $g^{(2)}(t_2 - t_1 = 0) > 1$, i.e. bunching of the

transmitted light. This bunching is expected as the subtraction operation reduces the mean photon number by $n_{sub} = 3$, without reducing the width of the photon number distribution compared to the incoming coherent pulses, thus inducing super-Poissonian statistics[15,48]. It also highlights that subtraction of exactly $n_{sub}$ photons is not to be confused with $n_{sub}$ consecutive applications of annihilation operators, which also reduces $\langle n_{in} \rangle$ of the incoming coherent pulses by up to $n_{sub}$, but maintains Poisson statistics, or probabilistic subtraction[49]. Furthermore, the interaction with the absorber induces correlations within the transmitted pulse for $t_2 \neq t_1$, the timescale of which can be observed on the anti-diagonal axes in Fig. 2c. The correlations disappear towards the end of the pulses due to two effects: as saturation suppresses photon absorption, the incoming coherent light is transmitted without change and, in addition, absorbers are more likely to have undergone random Raman decay, which introduces Poissonian fluctuations in the superatom state that ultimately become reflected in the photon statistics. Except for the early onset and end of the decay of the pulse, where the experimental signal is dominated by noise, we observe good agreement between theory and experiment.

To complement the transmission measurements, we also detect whether the absorbers are in the ground or a collective Rydberg state after the probe pulse by field ionisation of atoms in $|r\rangle$. By resolving the time-of-flight from the atomic clouds to detection on a multi-channel plate (MCP) (Fig. 1a), we can determine from which absorbers the produced ions originate. Figure 3a shows mean detected ions $\langle n_{ion} \rangle$ per pulse and absorber for $n_{sub} = 3$. The number of detection events from each absorber saturates at the detection efficiency $\eta$ as expected if no more than one excitation is supported. The slight deviations in $\langle n_{ion} \rangle$ for the individual absorbers result from a slight dependence of $\eta$ on the superatom position (between 0.18 and 0.25, 'Methods'). We also compare $\langle n_{ion} \rangle$ to the combined populations of $|W\rangle$ and $|D\rangle$ after the probe pulse in the three-level model, again in good agreement

with the experimental data following multiplication with the corresponding values of $\eta$. To verify that each absorber is saturated by exactly one excitation, and thus represents a single ion source[37,50], we analyse the counting statistics via the Mandel-$Q$ parameter $Q = \text{Var}(n_{ion})/\langle n_{ion} \rangle - 1$ (Fig. 3b), which gives $-\eta$ for perfect blockade (imperfect detection leads to a binomial distribution with success probability $\eta$), 0 for Poissonian, and $>0$ for super-Poissonian statistics. Analysing each absorber separately, we find $Q \approx -\eta$ for sufficient number of input photons, as expected for saturation at one, while analysis of the combined counts from all absorbers yields $Q \approx -\langle n_{ion} \rangle / n_{sub}$ as expected for saturation at $n_{sub}$ excitations.

Finally, Fig. 3c shows the ratio $Q/\langle n_{ion} \rangle$ for the three individual absorbers. For large input photon number $\langle n_{in} \rangle$ and perfect blockade, when each absorber contains exactly one excitation, this quantity should give $-1$. We observe a small deviation as $\langle n_{in} \rangle$ increases which indicates the possibility of additional Rydberg excitations beyond the number of absorbers. To account for these in the model results, we increase the excitation probabilities obtained from the Rydberg populations in the three-level model by a small, photon-number-dependent probability $p_2(n_{in})$, which is independent of the superatoms' states. In addition, we also account for the influence of dark counts in the ion detection by adding an offset independent of $\langle n_{in} \rangle$, based on the experimentally observed dark-count rate of 9 kHz. These fluctuations induce a small Poissonian component in the ion counting statistics and thus shift $Q/\langle n_{ion} \rangle$ to values above $-1$. The model results for $\langle n_{ion} \rangle$ and $\text{Var}(n_{ion})$ shown in Fig. 3c account for double Rydberg excitations and dark counts and reproduce the experimental results well for $p_2 = 3.5$, 6.5 and $5.0 \times 10^{-4}$ for the first, second and third absorber, respectively. To achieve good agreement for the second absorber, we need to increase the constant noise by a factor 5 compared to the dark-count rate. This is presumably due to the occasional detection of ions originating from the first and third absorber as well as atoms trapped in between the superatoms in the corresponding time-of-flight window attributed to the second absorber, which is expected to be more prone to these events due to its central position.

Our analysis supports the hypothesis that there is a small, $\langle n_{in} \rangle$-dependent probability to create additional Rydberg excitations, which may be caused by several mechanisms which we cannot distinguish in our experiment due to the small magnitude of the effect. First, residual atoms trapped between the superatom ensembles may be excited to $|r\rangle$ if $\langle n_{in} \rangle$ is sufficiently high so that power broadening becomes comparable to the AC-Stark shift induced by the tightly confining optical traps. Second, imperfections in the blockade can occur from interaction-induced pair-state resonances on shells within the blockaded volume[51], similar to the anti-blockade effect[52,53].

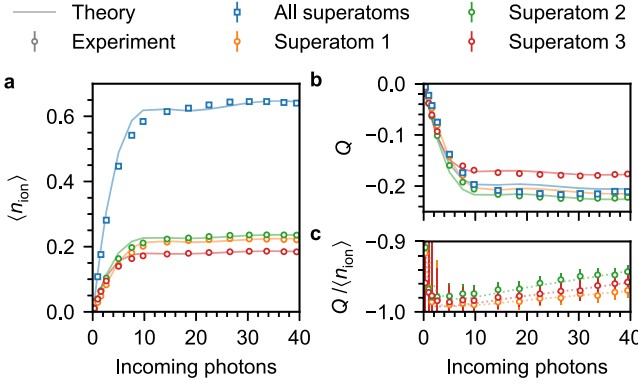

**Fig. 3 Probing the absorber states via field ionisation of atoms in $|r\rangle$.** The blue squares correspond to the combined ion signal from all three absorbers, while the yellow, green and red circles correspond to the statistics for the first, second and third absorber only. **a** Mean detected ions $\langle n_{ion} \rangle$ vs. mean incoming photons for $n_{sub} = 3$. The variation in the detected ions for the individual absorbers are the result of a position-dependent detection efficiency ('Methods'). The shaded line shows the Rydberg population predicted by the three-level model scaled by the corresponding detection efficiency. **b** Mandel-$Q$ parameter $Q$ for the same ion data as in (**a**). **c** Ratio $Q/\langle n_{ion} \rangle$ for the individual absorbers. The dotted line shows the results of the model with added noise due to double excitation and dark counts. Error bars correspond to one standard error and are smaller than the markers for most data points in (**a**) and (**b**).

**Parameter optimisation and scalability.** In the following theoretical analysis, we investigate the parameter space of the superatom photon absorber and discuss the potential for scaling beyond $n_{sub} = 3$. We base the discussion on the results of the Lindblad master equation for a one-dimensional chain of chirally coupled superatoms described by the three-level model[48,54,55] ('Methods'). On one hand, parameter optimisation is necessary as the Raman decay $\Gamma$ introduces an uncertainty about the number of absorbed photons and sets an upper limit on the pulse length $\tau$. For short $\tau$ on the other hand, $\gamma_D$ must be balanced with the driving strength $\sqrt{\kappa R_{in}}$ to yield high absorption probability.

We begin our analysis by considering stochastic loss of photons due to incoherent Raman decay from the Rydberg manifold into $|G\rangle$. It opens a scattering channel into non-observed modes, with a

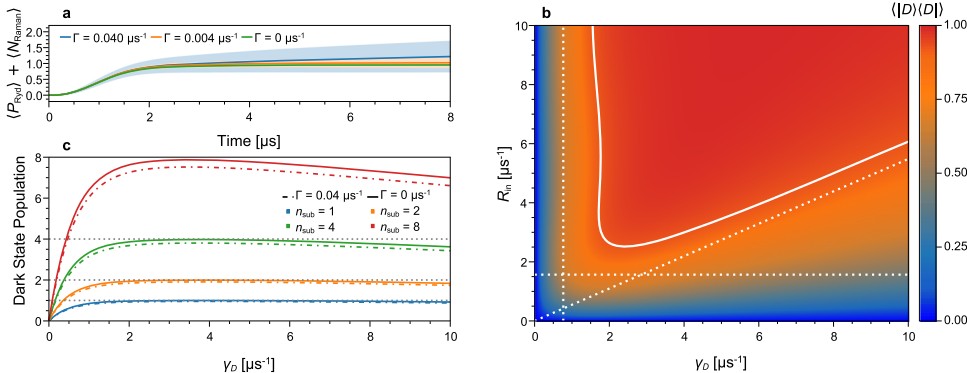

**Fig. 4 Theory prediction for the superatom dynamics.** All figures are at $\kappa = 0.35\ \mu s^{-1}$. **a** Total number of subtracted photons vs. time, i.e. the sum of the dark-state population and Raman emitted photons, for a single superatom at $R_{\rm in} = 5$ photons $\mu s^{-1}$ and dephasing rate $\gamma_D = 2.4\ \mu s^{-1}$. Shaded regions indicate one standard deviation of the number of Raman emitted photons. **b** Dark-state population, here equal to the photon absorption probability, after driving a single superatom for $\tau = 3\ \mu s$ without Raman decay $\Gamma = 0$. The solid line indicates the 90% level. The dashed horizontal, vertical and diagonal lines correspond to $\sqrt{\kappa R_{\rm in}}\tau = \pi/2$, $\exp(-\gamma_D\tau) = 0.1$ and $\exp(-4\kappa R_{\rm in}\tau/\gamma_D) = 0.1$, respectively. **c** Mean dark-state population in a chain of 1, 2, 4 and 8 superatoms after driving the superatoms for $\tau = 4\ \mu s$ at a constant rate of $R_{\rm in} = 5$ photons $\mu s^{-1}$.

mean number of lost photons

$$\langle N_{\rm Raman}(t)\rangle = \Gamma \int_{t_0}^{t} dt'\ P_{\rm Ryd}(t'), \tag{1}$$

where $P_{\rm Ryd}(t)$ is the combined population of $|W\rangle$ and $|D\rangle$. Figure 4a shows the total number of absorbed photons $P_{\rm Ryd}(t) + \langle N_{\rm Raman}(t)\rangle$ together with one standard deviation $\sqrt{{\rm Var}(N_{\rm Raman}(t))}$ ('Methods') for $\Gamma = 0$, 0.004 and 0.04 $\mu s^{-1}$, for a single superatom at constant driving. The fluctuations in photon absorption increase over time, highlighting that deterministic photon subtraction requires both low $\Gamma$ and short duration $\tau$.

Consequently, we now analyse the dynamics of a single superatom at $\tau = 3\ \mu s$, similar to the experiment. Figure 4b shows the population of $|D\rangle$ vs. $\gamma_D$ and $R_{\rm in}$ indicating a large parameter regime where photon absorption occurs with high probability. This regime is bounded by three processes with independent timescales, which we indicate by dashed lines. Firstly, the superatom is excited into $|W\rangle$ with rate $\sqrt{\kappa R_{\rm in}}$ and the requirement $\sqrt{\kappa R_{\rm in}}\tau \gg 1$ gives a lower bound for the necessary photon rate. Similarly, $|W\rangle$ decays into $|D\rangle$ with rate $\gamma_D$ and thus the dark state will only be populated for $\gamma_D\tau \gg 1$. However, at a large $\gamma_D$, the superatom dynamics enter an overdamped regime in which we can adiabatically eliminate $|W\rangle$ ('Methods') and the effective absorption rate scales asymptotically as $\gamma_{\rm eff} \simeq 4\kappa R_{\rm in}/\gamma_D$. Therefore, we also require $\gamma_{\rm eff}\tau \gg 1$, limiting the maximal dephasing rate $\gamma_D$. This analysis is valid until we reach a large $\sqrt{R_{\rm in}\kappa}\tau$ where we expect that the blockade mechanism breaks down and the superatom starts to absorb more than one photon.

Lastly, we solve the master equation for a chain of up to $n_{\rm sub}$ driven superatoms. Figure 4c compares the total dark-state population for an ideal system with $\Gamma = 0\ \mu s^{-1}$ (solid lines) to $\Gamma = 0.04\ \mu s^{-1}$ (dashed lines) and shows the potential to extend our photon subtraction scheme up to $n_{\rm sub} = 8$. In the simulation, we drive the superatoms with a mean number of 20 photons, which indicates that our set-up can work well even when the number of photons becomes comparable to $n_{\rm sub}$. The dark-state population never reaches $n_{\rm sub}$ exactly, which is due to the short pulse duration $\tau$ considered here and, by decreasing the Raman decay rate, higher absorption probabilities can be achieved.

## Discussion

The theoretical analysis of our experimental results reveals that the main contributions to imperfections in the subtracted photon

number are two-fold and are not necessarily unique to our scheme. First, a finite lifetime of the saturated state leads to excess absorption and the random nature of a decay process like Raman decay in our system introduces a probabilistic component into an initially deterministic scheme. This applies to any scheme which employs excited or metastable states and the severity of the impact depends on the decay strength compared to the absorber–photon coupling and the pulse duration. While the loss through Raman decay may initially seem as a disadvantage of our implementation, it should be noted that insertion loss into waveguides and cavities can lead to similar probabilistic fluctuations. Second, the slight deviation from $n_{\rm sub}$ for coherent input pulses of finite duration is more general and affects all subtraction schemes relying on irreversible transfer into a dark state or separate optical modes irrespective of the absorber nature, also including hybrid systems of waveguide- and cavity-coupled single quantum emitters[14,56].

In our scheme, Raman decay could be further suppressed by reducing either $\Omega_c$ or increasing $\Delta$, with the latter also reducing residual absorption on the probe transition. To compensate the associated reduction in $\kappa$, one can increase the number of atoms $N$ per superatom via the ensemble density or increase the probe waist with $r_B$ as upper constraint. Meanwhile, for increasing $\kappa$ combined with fine-tuning of $\gamma_D$, the dark-state population converges towards $n_{\rm sub}$ shifting the curves in Fig. 4c upwards. In this context, performance limitations will ultimately occur for high $R_{\rm in}$ as power broadening causes a breakdown of the blockade.

While high-fidelity preparation of quantum states of light may require more substantial performance improvements, limitations are less stringent for other applications of our set-up. A more readily implementable application is number-resolved detection of up to $n_{\rm sub}$ photons based on the number of absorbers in a Rydberg state. Currently, performance would be limited by the low-efficiency $\eta$ to detect the superatom state via field ionisation, but this could be significantly improved by replacing the MCP by another model or using optical detection[57–59]. By increasing $n_{\rm sub}$ well beyond the expected photon number, a weak photon–absorber coupling $\kappa$ could also be compensated. Meanwhile, it is still important to minimise Raman decay as it reduces the detection efficiency for each absorber.

In summary, we have experimentally demonstrated controlled multi-photon subtraction from weak coherent probe pulses using a cascaded chain of saturable Rydberg superatom absorbers in free space. Our theoretical analysis has identified both technical

and fundamental sources of imperfections, including the introduction of probabilistic fluctuations through decay into non-observed modes resulting in a probability slightly below unity to transfer an absorber into its saturated state for coherent input pulses of finite duration.

Obvious next steps include improving the subtraction fidelity via the measures discussed above, changes to the optical trapping of the atomic ensembles to further increase $n_{sub}$, and the implementation of optical readout for number-resolved photon detection. More generally, our system of cascaded superatoms is also well suited to study the behaviour of emitters coupled to a chiral waveguide[54,60,61], as the superatoms not only introduce photon correlations, but the photons also coherently mediate interactions between the superatoms, which should become evident in the limit $\gamma_D \ll \kappa$ and with increasing $n_{sub}$.

## Methods

### Ensemble preparation

We start from a cigar-shaped ensemble of $^{87}$Rb atoms in a crossed optical dipole trap (wavelength 1064 nm, $1/e^2$-waist $\approx 55 \, \mu m$, intersection angle 30°) loaded from a magneto-optical trap (MOT). Following a final compression of the MOT, the atoms are evaporatively cooled as we reduce the trap light intensity in two stages. For additional cooling and to reduce atom loss, we employ Raman sideband cooling for 16 ms during each of the linear evaporation ramps. To create multiple ensembles for multiple superatom absorbers, we generate multiple, tightly focused optical traps with an elliptical cross-section (wavelength 805 nm, $1/e^2$-waists $\approx 9 \, \mu m$ along and $\approx 29 \, \mu m$ perpendicular to the probe) that intersect perpendicularly with the cigar-shaped ensemble as well as the probe and control beams by feeding several RF signals into an AOD (as shown in Fig. 1 and discussed in 'Results'). An objective system translates the resulting differences in diffraction angle into different trap positions that can be tuned via the signals' frequencies over a range of order 100 $\mu m$, which is limited by the axial extend of the crossed region of the reservoir trap. In our experiments the separation between the ensemble centres is $\approx 75 \, \mu m$ for $n_{sub} = 2$ and $\approx 50 \, \mu m$ for $n_{sub} = 3$.

Before experiments, we ramp the crossed dipole trap intensity to zero to release atoms confined between the dimples before increasing it again to provide confinement in the radial probe direction for the superatom ensembles. In combination with the $1/e^2$-waist radius of the probe ($\approx 6.5 \, \mu m$), the dimple confinement restricts the excitation volume below the blockade range. The focus of the control beam is larger ($\approx 14 \, \mu m$) to limit the variation of $\Omega_c$ across the excitation volume.

### Experimental sequence

Following the preparation outlined above, we turn the crossed dipole trap off for 14 $\mu s$ every 100 $\mu s$, while the dimple traps are left on to maintain confinement along the probe direction. The resulting AC-Stark shift is compensated by adjusting the probe frequency accordingly and we ensure that all superatom absorbers have the same resonance frequency by individual fine-tuning via the power for each RF signal applied to the AOD. Besides the axial confinement, the AC-Stark shift also helps to suppress Rydberg excitation of residual atoms trapped in between the dimple potentials. Following each single experimental shot, we field-ionise any Rydberg atoms to gain information about the absorber state and avoid the presence of residual Rydberg excitations during the next iteration of the superatom excitation. The ions are detected on a MCP. In total, we repeat the cycle described above 500 times before releasing the atoms to obtain reference pulses of the probe light in the same manner in the absence of any atoms and preparing new atomic ensembles.

### Site-resolved ion detection

In order to attribute the ions detected following each experimental shot to Rydberg excitations in different ensembles, we use a time-of-flight method. We find that the pulses generated by the MCP detector following detection of an ion occur in a time window with a width of $\approx 30$ ns and a typical separation of several 10 ns between the arrival times for two superatom ensembles separated by 50 $\mu m$. Combined with the 3 ns time resolution of our data acquisition, this allows us to attribute a detection windows of 75 ns to the location of each individual superatom (with a 40 ns gap between the windows for the superatoms furthest from the detector). In Fig. 3, the detection efficiency varies between $\eta \approx 0.18$ and 0.25 depending on the position of a superatom and is generally highest in the single absorber case. The variation is caused by a grid of steel wires, which is placed in front of the detector to shield the atoms in the experimental region from the strong electric field produced by the MCP front plate and partially obstructs the ion trajectories. For a single superatom, the applied ionisation and steering fields can be adapted to minimise the influence of the grid, but for multiple absorbers we cannot avoid that a fraction of ions is blocked by the wires, which depends on the location of their origin.

### Theoretical description

We describe each superatom $i$ as an effective three-level atom whose ground state $|G_i\rangle$ is coupled to a collective excited state, the bright

state, $|W_i\rangle$ by the coherent probe field $\alpha(t)$ (with $R_{in} = |\alpha(t)|^2$). The photon absorber relies on shelving excitations into a non-radiating dark state $|D_i\rangle$, which we model by an incoherent decay of the bright state with rate $\gamma_D$. Assuming the dipole and rotating wave approximation and no Raman decay, the superatoms obey the master equation[54]

$$\partial_t \rho = -\frac{i}{\hbar}[H_{drive}(t) + H_{exc}, \rho] + \kappa \mathcal{D}\left[\sum_{i=1}^{N} \sigma_{W_i}^-\right]\rho \\ + \gamma_D \sum_{i=1}^{N} \mathcal{D}[\sigma_{D_i}^+ \sigma_{W_i}^-]\rho. \quad (2)$$

The master equation consists of the action of the probe field

$$H_{drive} = \sqrt{\kappa} \sum_{i=1}^{N} \left( \alpha(t)\sigma_{W_i}^+ + \alpha^*(t)\sigma_{W_i}^- \right), \quad (3)$$

a hopping term due to the exchange of virtual photons

$$H_{exc} = -\frac{i\kappa}{2} \sum_{i>j} \left( \sigma_{W_i}^+ \sigma_{W_j}^- - \sigma_{W_j}^+ \sigma_{W_i}^- \right) \quad (4)$$

and the dissipative decay terms $\mathcal{D}\left[\sum_{i=1}^{N} \sigma_{W_i}^-\right]$ and $\sum_{i=1}^{N} \mathcal{D}[\sigma_{D_i}^+ \sigma_{W_i}^-]$, describing the collective decay of the superatoms and dephasing of each bright state into the respective dark state. We use the notation $\sigma_{A_i}^- \equiv |G_i\rangle\langle A_i|$, $\sigma_{A_i}^+ \equiv |A_i\rangle\langle G_i|$ and $\mathcal{D}[\sigma]\rho \equiv \sigma\rho\sigma^\dagger - \{\sigma^\dagger\sigma, \rho\}/2$.

To understand non-deterministic effects in our photon absorber set-up, it is essential to include Raman decay in our model. In the simplest description, Raman decay enters as an additional decay term $\Gamma(\mathcal{D}[\sigma_{W_i}^-]\rho + \mathcal{D}[\sigma_{D_i}^-]\rho)$ for each superatom. To gain access to the number statistics of the emitted Raman photons, we further introduce a virtual spin chain to our model and modify the Raman decay terms so that each Raman decay yields an excitation in the spin chain. This allows us to calculate the standard deviation of the number of emitted Raman photons, as depicted in Fig. 4a.

At a large $\gamma_D$, the superatoms enter an overdamped regime and further increasing $\gamma_D$ negatively impacts the photon absorption rate. In the overdamped dynamics, we may adiabatically eliminate the bright state from the master equation, which we achieve by setting $\partial_t \rho_{WW} = 0 = \partial_t \rho_{WG} = (\partial_t \rho_{GW})^*$, with the shorthand notation $\langle A|\rho|B\rangle = \rho_{AB}$. Under the adiabatic elimination, the master equation for a single superatom reduces to a classical rate equation for the ground-state and dark-state population

$$\partial_t \rho_{GG} = -\gamma_{eff} \rho_{GG} + \Gamma \rho_{DD}, \quad (5)$$

$$\partial_t \rho_{DD} = \gamma_{eff} \rho_{GG} - \Gamma \rho_{DD}, \quad (6)$$

where the effective decay rate reads as

$$\gamma_{eff} = \frac{4\kappa R_{in}\gamma_D}{(\kappa + \Gamma + \gamma_D)^2 + 4\kappa R_{in}}. \quad (7)$$

## Data availability

The data generated in this study have been deposited in the Zenodo database under accession code https://doi.org/10.5281/zenodo.4984099

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

## Acknowledgements

This work has received funding from the European Union's Horizon 2020 programme under the ERC consolidator grants SIRPOL (grant no. 681208) and RYD-QNLO (grant no. 771417), the ErBeStA project (grant no. 800942), and under grant agreement no. 845218 (Marie Skłodowska-Curie Individual Fellowship to H.B.), the Deutsche For-schungsgemeinschaft (DFG) under SPP 1929 GiRyd project BU 2247/4-1, and the Carlsberg Foundation through the Semper Ardens Research Project QCooL.

## Author contributions

N.S., H.B., M.G.H. and S.H. performed the experimental work, K.K., J.K. and H.P.B. conducted the theoretical analysis. All authors contributed to data analysis, discussion of the results, and the preparation of the manuscript.

## Funding

## Competing interests

The authors declare no competing interests.
