## [Peer Review File · Nature Communications]

Reviewers' Comments:

Reviewer #1:

Remarks to the Author:

In this work, Stiesdal and co-workers use up to three Rydberg superatoms to subtract up to three photons from laser pulses. The mechanism relies on the Rydberg blockade effect. They investigate different effects caused by the Rydberg superatom absorbers on the laser pulses, including the reduction of the laser pulse energy, the distortion of the pulse shapes and photon-bunching. These investigations are conducted using different numbers of absorbers and different laser pulse energies. In addition, they probe the statistics of the Rydberg atoms produced by detecting the Rydberg atoms using field ionisation. Further, they model their system and identify the key parameters which affect the efficiency of the photon subtraction process. Photon subtraction has applications in quantum metrology, quantum computation and simulation, and photon-number resolving measurements.

The experimental results are convincing. The methodology and analysis are very similar to an earlier work from the same group (Ref 35), in which a single photon was subtracted using a single Rydberg superatom. Increase in system size to three superatoms is an important step towards applications in quantum metrology, quantum computation and simulation, as well as photon-number-resolved detection, and so I believe it should warrant publication in Nature Communications, once some issues are addressed.

1. The abstract refers to "engineered dephasing", however the main text gives no indication that γ_D was engineered. I think it would be appropriate to mention this, and describe how the engineering was conducted. This might be introduced around the bottom of the second column on page 2, where the contributions to γ_D are discussed.
2. In the abstract "We show that our scheme should scale well to higher absorber numbers", where is this shown? Scaling-up suggests increasing a system size by orders of magnitude (or at least one order of magnitude). Fig 4c shows simulation results for only eight absorbers.
3. The introduction mentions using photon subtraction to achieve quantum metrology, and cites ref. 6. In the cited article, the authors consider a probabilistic subtraction scheme, and they say that when the success probability is considered, any quantum-based metrology enhancement is lost. With the Rydberg-superatom subtraction scheme, can the quantum-based metrology enhancement be preserved, or will it be ultimately lost?
4. Reference 8 describes experimental characterisation of the annihilation operator, not the subtraction operator, why is it cited?
5. In the introduction, "we analyse the performance of our system and find that Raman decay is the main source of deviations from the ideal absorber behaviour." I don't see any such analysis in the manuscript.
6. In the introduction, "scaling beyond $n_{\text{sub}}=3$... is realistic as long as the decay can be further suppressed", up until this point there has been no mention of your efforts to suppress Raman decay, so perhaps it is more appropriate to simply write "scaling beyond $n_{\text{sub}}=3$... is realistic as long as Raman decay can be suppressed."
7. It wasn't initially clear to me whether the "axial extent of the ensemble" referred to the extent of the ensemble along the direction of the probe beam's propagation, or to the long-axis of the cigar-shaped ensemble. I think the text in the final paragraph of page 1 might be clearer if it was changed along the lines: "If the probe beam waist ($1/e^{*2}$...) and the extent of the ensemble along the probe beam axis constrain the excitation volume...", similarly at the top of column 2 on page 2 instead of "axially confine each ensemble", "confine each ensemble along the probe beam axis" and instead of "with an axial extent $< r_B$ ", "with an extent $< r_B$ along the probe beam axis".
8. At the bottom of the second column on page 1, κ is introduced but it isn't defined. Also, I think it would help to write that g_0 is the single-probe-photon-single-atom coupling strength for the $|g\rangle \leftrightarrow |e\rangle$ transition, and that $g_0 \Omega_c / (2 \Delta)$ is the effective single-probe-photon-single-atom coupling strength for the $|g\rangle \leftrightarrow |r\rangle$ transition.
9. In Fig 1b, it isn't clear which of the upper two levels is the shifted by the vdW interaction V . If a

reader interprets the upper level as being shifted by V , then the reader will wrongly interpret that the system is operated in the anti-blockade/facilitation regime. I think V shouldn't be shown in a single-atom level scheme, but rather only in level schemes with two-atom states gg , gr , and rr or multi-atom states.

10. In Fig 1c the decays with rates Γ and γ_D should have squiggly lines, the text $\sqrt{\kappa R}$ should be closer to the red arrow, and the red arrow should be double-sided, to indicate that $|W\rangle \rightarrow |G\rangle$ can be driven (hence the need for γ_D). Also, the arrows in Fig 1b should be doubled-sided.

11. In Fig 1a the four single-photon counters are shown. Why are four used, instead of two? The fact that there are four is not mentioned in the main text or in the Methods section. To calculate the g_2 function, I expect you follow the same methodology as in your earlier work (reference 35) "we calculate the correlation between all distinct pairs of our four counters and average over these results", I think repeating this text in this manuscript would be useful.

12. At the top of the second column on page 2, instead of $N \propto 10^4$, should a " \sim " be used? And the text "depending on n_{abs} ", does this mean that when $n_{\text{abs}}=3$ that N was 3 times smaller than when $n_{\text{abs}}=1$? This could be written more clearly.

13. Do n_{sub} and n_{abs} both refer to the number of absorbers? Then only one of them should be used.

14. Second column, page 2, in the absence of the control beam, the transmission of the probe beam is $>99\%$. This value is compared with the insertion losses of optical fibres, apparently signally a benefit of the "free-space" Rydberg superatom subtraction scheme. However, the experimental setup is not completely "free-space", there is a single-mode fibre between the absorbers and the detectors. How important is the single-mode fibre in the setup?

15. In the second column of page 2, "the pulse area $\sqrt{\kappa R} \tau$ does not reach π as required to fully drive the superatom to $|W\rangle$ and $|D\rangle$ ". If the pulse area was π , the superatom would not be fully driven to $|W\rangle$, due to γ_D , full transfer is only obtained in the limit pulse area \rightarrow infinity (if one can neglect Raman decay).

16. In the second column of page 2, "to suppress superradiant reemission of absorbed photons in the forward direction after the probe pulse". Superradiant reemission of absorbed photons occurs not after the probe pulse, but rather during the probe pulse.

17. In the second column of page 2, "The dephasing is dominated by ... the AC-Stark shift induced by the trapping light." In the Methods, Experimental sequence, compensation of the AC-Stark shift is described. Do I understand correctly, that the AC-Stark shift is on average compensated, but that the difference of AC-Stark shifts experienced by different atoms which experience different trapping field strengths gives rise to the dephasing? If so, I suggest changing the text in the second column of page 2 to reflect this.

18. Top of the first column of page 3: why do κ , Γ and γ_D vary so much as n_{sub} is varied? In particular, is the variation of Γ consistent with what one expects from Raman decay? Was Ω_c or Δ varied? What are the uncertainties on the estimates of κ , Γ and γ_D ?

19. In the first column of page 3, n_{in} is introduced but not defined.

20. In the first column of page 3, "re-emission occurs back into the probe mode in forward direction with rate κ ". Isn't the rate the coupling rate $\sqrt{\kappa R_{\text{in}}}$?

21. In the experimental data, e.g. in Fig 2a, how large is the uncertainty in the incoming/outgoing photon number, due to uncertainty in the viewport transmission, power meter calibration, unfiltered control photons, uncertainty in the fibre coupling efficiency etc?

22. In the first column of page 3, the absolute symbol can be removed from $g_2(|t_2-t_1|)$.

23. Fig 2b the fitted model is not visible, I suggest using a clear dashed line of a different colour on top of the experimental data.

24. Fig 2b and 2c, why does the time axis go beyond 3.5us? Is there photon counts in the top-right graph in Fig 2b at times $>3.5\mu\text{s}$? In the simulation results I expect that for $t > 3.5\mu\text{s}$ the count rate should be zero, so then the g_2 function should not be well-defined, and so it is better not to show any simulation results for $t > 3.5\mu\text{s}$.

25. The color plots in 2c are quite pretty, but I think they should be replaced by simple 2D graphs

showing only the results from along the diagonal ($t_1=t_2$) [i.e. with x-axis time= $t_1=t_2$, and y-axis $g(2)$], because the results along the diagonal are discussed in the paper, whereas the off-diagonal results are not discussed. A simple 2D graph would allow better comparison between the theory and the experimental results.

26. Why is $g_2 < 1$ for $t_1=t_2 < 1\mu\text{s}$? In the main text it says that "the experimental signal is dominated by noise" during the rise-time. However, the simulations also show $g_2 < 1$ for $t_1=t_2 < 1\mu\text{s}$.

27. In the second column of page 3, I don't understand the second reason for the disappearance of the correlations towards the end of the pulses, "in addition absorbers are more likely to have undergone random Raman decay...", can you explain it further to me please? Does it cause a "washing-out" of any correlations?

28. In the second column of page 3, regarding "the total number of ions saturates at the sum of the three absorbers", I understand this to mean that in Fig 3a the sum of the orange, green and red curves is the same as the blue curve. Why is this written here? I understand that depending on the time the ion reaches the MCP, the ion is interpreted as having come from either the first, second or third absorber. The detection windows have widths of 75ns (Methods). Are there gaps between the detection windows? If there are no gaps between them, then it seems obvious that the sum of the three curves will equal the blue curve. Furthermore, what is the motivation behind showing the blue curve in Fig 3a? I think that by removing the blue curve, the y-range will be reduced, and it will be easier to see how flat $\langle n_{\text{ion}} \rangle$ is for incoming photons > 10 .

29. In the second column of page 3, in the formula for the Mandel-Q parameter, I think adding parentheses would aid readability $\text{Var}(n_{\text{ion}})$...

30. In the second column of page 3, after the formula for the Mandel-Q parameter, I think adding "which gives $-\eta$ for perfect blockade (imperfect detection leads to a binomial distribution with success probability η)" will make it easier for the reader to understand where $Q = -\eta$ comes from.

31. In the caption of Fig 3, I think the word "scaled" is more appropriate than "normalised".

32. First column of page 4, I don't think the phrasing "We account for these in the theoretical analysis..." is appropriate, I think it's better to say write "We account for this in the model by increasing the Rydberg populations by a small photon-number dependent probability...". The reason is that the curves in Fig 3c do not show the theoretical analysis, rather they show the results given by the model. The theoretical analysis involves the reasoning that there can be possible additional Rydberg excitations, etc.

33. First column of page 4, the deviation of $Q/\langle n_{\text{ion}} \rangle$ from -1 is caused by double Rydberg excitation and dark counts. The double Rydberg excitation seems to be visible in Fig 3a, I think it's easiest for the reader to digest if the double Rydberg excitation is already introduced earlier when the results of Fig 3a are discussed.

34. First column of page 4, "The results based on the modified values for $\langle n_{\text{ion}} \rangle$ and $\text{Var} n_{\text{ion}}$ are shown...", at first, I read this as if the experimentally-measured values $\langle n_{\text{ion}} \rangle$ and $\text{Var} n_{\text{ion}}$ had been modified before being plotted in Fig 3c, whereas in actuality the model was modified to include effects of double Rydberg excitations and dark counts. I think changing the text to something along the lines of "The model curves shown in Fig 3c account for double Rydberg excitations and dark counts" would help.

35. First column, page 4, why does p_2 vary so much between the absorbers?

36. Second column, page 4, "Our analysis underpins the hypothesis", I don't think the word "underpins" is appropriate. The model uses the hypothesis, the model is compared with the experimental results, and the favourable comparison supports the hypothesis.

37. At the start of the section "Parameter optimisation and scalability", the text reads "we determine the optimal parameters for the superatom photon absorber". As far as I understand, the free parameters in the simplified three-level model are κR_{in} , τ and γD . A true optimisation would involve a three-dimensional scan of all these parameters, however this is not done here. The text should be changed to reflect this. I expect that if one was to find optimal parameters using this model, one would obtain $\kappa R_{\text{in}} \rightarrow \text{infinity}$.

I think this section clearly illustrates what are the main competing factors are towards achieving high-efficiency subtraction process.

38. Second column, page 4, "the Raman decay introduces a probabilistic component into the

otherwise deterministic scheme". Doesn't the dephasing γ_D also introduce a probabilistic component to the scheme?

39. Second column, page 4, "This regime is bounded by three processes with independent time scales, which we indicate by dashed lines." I think it's worth mentioning that when the photon rate/ κ are increased too far, then second Rydberg excitations will not be blocked, and the model breaks down.

40. Fig 4a and 4c the legend text should be changed to μs to match the rest of the text. In Fig 4b the colorbar axis is missing a label, the y-label "Photons" suggests a number, while it is a rate, and should be changed to " R_{in} ". The legend in Fig 4c is missing information about the different coloured curves. I think that rescaling the y-axis in Fig 4c from Dark State Population \rightarrow Dark State Population / n_{sub} would help. Right now, it is very difficult to compare the curves.

41. I understand the number of subtracted photons to be $\langle P_{Ryd} \rangle + \langle N_{Raman} \rangle$, however the caption of Fig 4 suggests that the number of subtracted photons equals $\langle P_{Ryd} \rangle$. Right now, the caption text does not explain Fig 4 well. To understand this figure, the reader needs to read the main text. The caption text should be developed.

42. First column of page 5, I suggest changing the text from the idealistic "in the absence of Raman decay, the absorption probability can be made arbitrarily large by increasing τ " to something more physical "by decreasing the Raman decay rate, higher absorption probabilities can be achieved."

43. Second column, page 5, "An immediate application is number-resolved detection of up to n_{sub} photons based on the number of absorbers in a Rydberg state ... by increasing n_{sub} well beyond the expected photon number, a weak photon-absorber coupling κ could be compensated". It would be nice if there was data to support the "immediate" applicability of the method. However, Fig 4b doesn't support this, given that the pulse considered involved 20 photons, which is greater than the number of absorbers.

I think a reader would like to know how many absorbers would be needed to resolve the number of photons in a pulse of up to ~ 5 photons with a fidelity of $\sim 90\%$ (with reasonable experimental parameters)? I would like to see a 2D colour graph with n_{sub} on the x-axis, the number of photons on the y-axis, and the colour indicating the expected fidelity (or infidelity) of a photon-number-resolving measurement. I expect that a photon-number-resolving measurement device using an array of Rydberg superatoms would have difficulty out-performing a bunch of beam splitters and single-photon detectors.

44. Second column, page 5, the detection efficiency "could be significantly improved by replacing the MCP ... or using optical detection". How far is it reasonable to expect that the detection efficiency can be improved?

45. Second column, page 5, "Raman decay should still be strongly suppressed", it isn't clear when reading this whether "should" is meant in a passive or active sense. Something along the lines of "A high detection efficiency also requires suppressing Raman decay, which acts to reduce the efficiency." Might be better.

46. First column, page 6, "in principle deterministic scheme", as I mentioned earlier, I think both Raman decay and the engineered dephasing make the scheme non-deterministic.

47. Methods, second column page 6, do you know whether the width of the time window due to jitter of the MCP or the distribution of the ion flight times?

Reviewer #2:

Remarks to the Author:

The authors report a study of sequential single-photon subtraction by Rydberg superatoms. The central effect of the study - single-photon subtraction - is expected to increase the value of g_2 at small delays, and the authors measure such an effect in a system of three superatoms. The experimental results are convincing and supported by extensive theoretical analysis. The technique they demonstrate can be further refined and applied for more involved manipulations of non-classical states of light and atoms. I recommend the manuscript for publication in its present form.

Reviewer #3:

Remarks to the Author:

The authors report an experimental and theoretical study of controlled multi-photon subtraction with cascaded Rydberg super atoms.

Their scheme exploits the Rydberg blockade in small atomic ensembles, which means that merely one atom out of many can be excited to a Rydberg state. This leads to a delocalised Rydberg excitation, which dephases quickly. This mechanism strongly suppresses the reemission of the absorbed photon into the mode of the excitation beam. This realises a single photon absorber, and the authors investigate and characterise a system that is composed of three such concatenated absorbers.

The work is elegant, timely and interesting - especially when considering that such passive devices may find applications in optical quantum information processing and communication. The theory strongly supports the experimental findings and a serious account of possible error sources and performance limitations is given. The associated discussion is not only relevant for the particular system of Rydberg superatoms, but highlights general issues when building quantum devices that are based on few-level systems subject to decoherence.

I do not see any shortcoming of this work and - due to the timeliness, relevance and high quality of the results - I recommend publication in Nature Communications.

Further comments:

- 1) The dashed lines in the inset of Fig 2a are too faint.
- 2) I think it would be good to rethink the presentation of the data in panel 2b. It is very hard to discriminate the two colored curves that are overlaid in each sub-panel.
- 3) It might be worth pointing out that the excitation of a Rydberg superatom together with ionisation realises a single ion source, e.g. discussed in [Physical Review Letters 110, 213003 (2013)].

Referee 1

In this work, Stiesdal and co-workers use up to three Rydberg superatoms to subtract up to three photons from laser pulses. The mechanism relies on the Rydberg blockade effect. They investigate different effects caused by the Rydberg superatom absorbers on the laser pulses, including the reduction of the laser pulse energy, the distortion of the pulse shapes and photon-bunching. These investigations are conducted using different numbers of absorbers and different laser pulse energies. In addition, they probe the statistics of the Rydberg atoms produced by detecting the Rydberg atoms using field ionisation. Further, they model their system and identify the key parameters which affect the efficiency of the photon subtraction process. Photon subtraction has applications in quantum metrology, quantum computation and simulation, and photon-number resolving measurements.

The experimental results are convincing. The methodology and analysis are very similar to an earlier work from the same group (Ref 35), in which a single photon was subtracted using a single Rydberg superatom. Increase in system size to three superatoms is an important step towards applications in quantum metrology, quantum computation and simulation, as well as photon-number-resolved detection, and so I believe it should warrant publication in Nature Communications, once some issues are addressed.

Reply: We are very grateful for your careful and extensive review and overall positive feedback. The issues raised are addressed point-by-point below.

1. The abstract refers to “engineered dephasing”, however the main text gives no indication that γ_D was engineered. I think it would be appropriate to mention this

and describe how the engineering was conducted. This might be introduced around the bottom of the second column on page 2, where the contributions to γ_D are discussed.

Reply: The word engineered was mainly chosen to highlight that - unlike in many other experiments - the dephasing is desired and necessary to implement photon subtraction using Rydberg superatoms, but we agree that it suggests a much higher degree of control over the dephasing. Hence, we have changed the wording to “rapid”.

The main consideration in our experiment was to use co-propagating probe and control beams to increase motional dephasing compared to the more common counter-propagating configuration. We have added a brief statement on this in the paragraph suggested by the referee.

2. In the abstract “We show that our scheme should scale well to higher absorber numbers”, where is this shown? Scaling-up suggests increasing a system size by orders of magnitude (or at least one order of magnitude). Fig 4c shows simulation results for only eight absorbers.

Reply: We would of course have liked to perform simulations with > 8 absorbers, i.e. over 12 orders of magnitude, but were limited by memory requirements due to the scaling of the Hilbert space with n_{sub} . We have now rephrased the sentence to “...and investigate the performance of the multi-photon subtractor for increasing absorber numbers in the presence of Raman decay.”

3. The introduction mentions using photon subtraction to achieve quantum metrology and cites ref. 6. In the cited article, the authors consider a probabilistic subtraction scheme, and they say that when the success probability is considered, any quantum-based metrology enhancement is lost. With the Rydberg-superatom subtraction scheme, can the quantum-based metrology enhancement be preserved, or will it be ultimately lost?

Reply: Reference 6 describes the use of photon subtracted and photon added single mode states for quantum metrology. For this single mode assumption, the discussion in reference 6 is connected to our work only as an outlook; since the resulting photonic state in our setup is multi-modal with temporal and spatial correlations. The achievable quantum advantage of our photon states for metrology is still an open question and is subject to future research.

4. Reference 8 describes experimental characterisation of the annihilation operator, not the subtraction operator, why is it cited?

Reply: We thank the referee for this remark and have removed the citation.

5. In the introduction, “we analyse the performance of our system and find that Raman decay is the main source of deviations from the ideal absorber behaviour.” I don’t see any such analysis in the manuscript.

Reply: Here, we were referring to the section “Parameter optimisation and scalability”, i.e. the results presented in Fig. 4a, c which clearly shows the detrimental effect of increasing

Raman decay. We have reworded the sentence, so this becomes more clear now in conjunction with the following sentence.

6. In the introduction, “scaling beyond $n_{\text{sub}}=3$... is realistic as long as the decay can be further suppressed”, up until this point there has been no mention of your efforts to suppress Raman decay, so perhaps it is more appropriate to simply write “scaling beyond $n_{\text{sub}}=3$... is realistic as long as Raman decay can be suppressed.”

Reply: We agree that the statement could be confusing and have reworded it as suggested.

7. It wasn't initially clear to me whether the “axial extent of the ensemble” referred to the extent of the ensemble along the direction of the probe beam's propagation, or to the long-axis of the cigar-shaped ensemble. I think the text in the final paragraph of page 1 might be clearer if it was changed along the lines: “If the probe beam waist ($1/e^{**2}$...) and the extent of the ensemble along the probe beam axis constrain the excitation volume...”, similarly at the top of column 2 on page 2 instead of “axially confine each ensemble”, “confine each ensemble along the probe beam axis” and instead of “with an axial extent $< r_B$ ”, “with an extent $< r_B$ along the probe beam axis”.

Reply: We have reworded both statements as suggested.

8. At the bottom of the second column on page 1, κ is introduced but it isn't defined. Also, I think it would help to write that g_0 is the single-probe-photon-single-atom coupling strength for the $|g\rangle \leftrightarrow |e\rangle$ transition, and that $g_0 \Omega_c / (2 \Delta)$ is the effective single-probe-photon-single-atom coupling strength for the $|g\rangle \leftrightarrow |r\rangle$ transition.

Reply: We have added the additional definitions mentioned.

9. In Fig 1b, it isn't clear which of the upper two levels is the shifted by the vdW interaction V . If a reader interprets the upper level as being shifted by V , then the reader will wrongly interpret that the system is operated in the anti-blockade/facilitation regime. I think V shouldn't be shown in a single-atom level scheme, but rather only in level schemes with two-atom states gg , gr , and rr or multi-atom states.

Reply: We agree that showing V in a single-atom level scheme can be misleading. We have replaced it with the two photon detuning δ and have slightly reworded the main text to match the revised figure.

0. In Fig 1c the decays with rates Γ and γ_D should have squiggly lines, the text $\sqrt{\kappa R}$ should be closer to the red arrow, and the red arrow should be double-sided, to indicate that $|W\rangle \rightarrow |G\rangle$ can be driven (hence the need for γ_D). Also, the arrows in Fig 1b should be doubled-sided.

Reply: We have partially adapted the figure along the lines suggested by the referee, though have opted for dashed rather than squiggly lines for the dephasing and decay.

11. In Fig 1a the four single-photon counters are shown. Why are four used, instead of two? The fact that there are four is not mentioned in the main text or in the Methods section. To calculate the g_2 function, I expect you follow the same methodology as in your earlier work (reference 35) “we calculate the correlation between all distinct pairs of our four counters and average over these results”, I think repeating this text in this manuscript would be useful.

Reply: The referee is correct that we employ the same methodology as in ref. 34 and we agree that we should mention this in the text. The reason that our setup uses four counters is to be able to investigate higher-order correlations, which are not investigated in this work. To clarify the methodology we added the following sentence as suggested: “This configuration allows us to analyse second-order photon correlations by calculating the correlation between all distinct pairs and averaging over the results.”

12. At the top of the second column on page 2, instead of $N \propto 10^4$, should a “~” be used? And the text “depending on n_{abs} ”, does this mean that when $n_{\text{abs}}=3$ that N was 3 times smaller than when $n_{\text{abs}}=1$? This could be written more clearly.

Reply: We have changed the symbol as suggested. The dependence on n_{sub} is indeed the result of sharing the atoms initially trapped in our reservoir trap between a larger number of dimple traps (see Methods). However N is not exactly 3 times smaller as a larger number of dimple traps also leads to more atoms being transferred from the reservoir into the dimples all together. This is highlighted by the fact that the optical depth experienced by the probe (11, 16, and 20 for 1, 2 and 3 ensembles respectively) is not constant over n_{sub} .

13. Do n_{sub} and n_{abs} both refer to the number of absorbers? Then only one of them should be used.

Reply: This is correct, both n_{sub} and n_{abs} refer to the number of absorbers, we overlooked some remaining occurrences of n_{abs} when changing the notation to n_{sub} during preparation of the manuscript and thank the referee for pointing these out.

14. Second column, page 2, in the absence of the control beam, the transmission of the probe beam is >99%. This value is compared with the insertion losses of optical fibres, apparently signally a benefit of the “free-space” Rydberg superatom subtraction scheme. However, the experimental setup is not completely “free-space”, there is a single-mode fibre between the absorbers and the detectors. How important is the single-mode fibre in the setup?

Reply: We agree that this point might be less strong than suggested by our statement as there is of course coupling loss from a) coupling the initial probe pulse into a single-mode fibre before the experiment and b) coupling the transmitted pulse into another single mode fibre. We have therefore removed the comparison as well as a statement on insertion loss from the introduction. We have also reworded the sentence on the probe transmission to make it clearer that the loss occurs due to off-resonant scattering of the probe on the $|g\rangle \rightarrow |e\rangle$ transition.

On the fibres: The single-mode fibre at the input defines the probe mode and any incoming pulses must thus be matched to the fibre input. If properly mode-matched, the fibre on the

output acts as a mode filter for the transmitted probe light. In theory, there is no fundamental reason to use a single mode fibre on the output (or input) as long as the input and output modes of the probe are matched and no other than the transmitted probe photons enter the mode. In practice, using a multimode fibre or free-space detector would lead to an unwanted background due to stray light and fluorescence from Raman decay of the superatoms.

15. In the second column of page 2, “the pulse area $\sqrt{\kappa R} \tau$ does not reach π as required to fully drive the superatom to $|W\rangle$ and $|D\rangle$ ”. If the pulse area was π , the superatom would not be fully driven to $|W\rangle$, due to γ_D , full transfer is only obtained in the limit pulse area \rightarrow infinity (if one can neglect Raman decay).

Reply: We thank the referee for pointing out this subtlety, which is of course also evident later in the paper. We have reworded the statement to “...the pulse area $\sqrt{\kappa R_{in}} \tau$ is insufficient to predominantly drive the superatom population into $|W\rangle$ and $|D\rangle$.”

16. In the second column of page 2, “to suppress superradiant reemission of absorbed photons in the forward direction after the probe pulse”. Superradiant reemission of absorbed photons occurs not after the probe pulse, but rather during the probe pulse.

Reply: We disagree with the referee’s statement that there is no superradiant emission following the pulse. Due to the collective nature of the system, superradiance will occur as long as some degree of coherence is left in the system. This has previously been observed in Ref.s 43 and 44 respectively, as well as for ensembles of two-level atoms in various references cited in those papers.

It is of course correct that there is also superradiant re-emission into the probe mode during the pulse. This is however less relevant as the corresponding photons are not removed from the probe mode and the superatom is not saturated such that it can still absorb another photon.

17. In the second column of page 2, “The dephasing is dominated by ... the AC-Stark shift induced by the trapping light.” In the Methods, Experimental sequence, compensation of the AC-Stark shift is described. Do I understand correctly that the AC-Stark shift is on average compensated, but that the difference of AC-Stark shifts experienced by different atoms which experience different trapping field strengths gives rise to the dephasing? If so, I suggest changing the text in the second column of page 2 to reflect this.

Reply: We have added a comment on this as suggested: “...non-uniform AC-Stark shift induced by the trapping light (Methods), which can only be compensated for on average.”

18. Top of the first column of page 3: why do κ , Γ and γ_D vary so much as n_{sub} is varied? In particular, is the variation of Γ consistent with what one expects from Raman decay? Was Ω_c or Δ varied? What are the uncertainties on the estimates of κ , Γ and γ_D ?

Reply: The parameters κ and Γ are influenced by parameters of the probe and control fields, and in the case of κ , also the atom number per ensemble. The variation in κ is mostly due to the difference in the latter as n_{sub} is varied (see answer 12). For Γ , we attribute the variation to imperfections and drift in the alignment of the control beam relative to the dimple traps, both in terms of their angle to each other and the focal positions. I.e., we observe lower decay for $n_{\text{sub}} = 2$ where only the two outer traps are used. This would be consistent with higher Ω_c for the dimple towards the centre, e.g., due to a slight angular deviation. We did not vary Ω_c or Δ in this work to investigate their influence but have done so in our previous work (Ref. 44).

Meanwhile, γ_D is not affected by the probe and control parameters and alignment, but rather atomic motion. The cooling dynamics as the atoms are evaporated from the reservoir trap into the dimples are rather complex and vary with n_{sub} . We therefore attribute the variation in γ_D to differences in the ensemble temperature and trap potentials. Unfortunately, limited imaging resolution of our setup perpendicular to the AOD/objective plane makes a quantitative study of these dynamics difficult.

It is also important to note that the values are estimated on the assumption that all absorbers are equal. Differences in the actual parameters lead to a washing-out of the features in the light transmission for $n_{\text{sub}} > 1$. These differences exceed the uncertainties one would obtain from a fit to the transmitted signal. For this reason, we do not specify any uncertainties in the parameters since they are meant to be estimates for the modelling rather than an exact description of the actual system. To avoid the impression that the parameters are an exact fit, we have reduced the number of significant digits to which we state them to two for all values.

19. In the first column of page 3, n_{in} is introduced but not defined.

Reply: We had not reintroduced the definition of n_{in} as it is already stated in the beginning of the section “Results”, i.e. “We measure the transmission for coherent, Tukey shaped probe pulses with ... and mean incoming photon numbers $n_{\text{in}} \leq 40$ using...”. We have slightly rephrased this to “... and $n_{\text{in}} \leq 40$, where $\langle n_{\text{in}} \rangle$ is the mean incoming photon number per pulse, using...” to make the definition clearer.

20. In the first column of page 3, “re-emission occurs back into the probe mode in forward direction with rate κ ”. Isn’t the rate the coupling rate $\sqrt{\kappa R_{\text{in}}}$?

Reply: We emphasised that we mean spontaneous decay, which happens with rate κ .

21. In the experimental data, e.g. in Fig 2a, how large is the uncertainty in the incoming/outgoing photon number, due to uncertainty in the viewport transmission, power meter calibration, unfiltered control photons, uncertainty in the fibre coupling efficiency etc?

Reply: We characterize the transmission loss at classical intensities with a linear power meter measuring before/after the vacuum chamber and before the detectors such that all the stated transmission and coupling losses are accounted for in our photon number calibration. Hence, the major uncertainty that remains is the systematic error due to the power meter calibration and accuracy, which is on the few % level.

22. In the first column of page 3, the absolute symbol can be removed from $g_2(|t_2-t_1|)$.

Reply: This is of course correct, and we have removed the absolute value as suggested.

23. Fig 2b the fitted model is not visible, I suggest using a clear dashed line of a different colour on top of the experimental data.

Reply: We agree and have modified the figure such that the theory data is shown in dark grey to add clarity. Please note, that we have used experimentally recorded reference probe pulses as the basis of the simulation which explain the noise on the theory data.

24. Fig 2b and 2c, why does the time axis go beyond 3.5us? Is there photon counts in the top-right graph in Fig 2b at times >3.5us? In the simulation results I expect that for $t > 3.5\mu\text{s}$ the count rate should be zero, so then the g_2 function should not be well-defined, and so it is better not to show any simulation results for $t > 3.5\mu\text{s}$.

Reply: We have changed the time axes such that no data beyond 3.7 μs is shown. We note however, that there is a small photon signal after the pulse, also in the theory results. A superatom can re-emit a photon into the original probe mode as long as it has not fully dephased into $|D\rangle$ (see also answer 20).

25. The color plots in 2c are quite pretty, but I think they should be replaced by simple 2D graphs showing only the results from along the diagonal ($t_1=t_2$) [i.e. with x-axis $\text{time}=t_1=t_2$, and y-axis $g(2)$], because the results along the diagonal are discussed in the paper, whereas the off-diagonal results are not discussed. A simple 2D graph would allow better comparison between the theory and the experimental results.

Reply: We have discussed the presentation of figure 2c amongst all authors and have concluded that we prefer to keep the current presentation for two reasons: First, unlike a 1D plot, the 2D plot shows that the interaction with the absorber also introduces correlations for $t_2 \neq t_1$. The characteristic timescale of these becomes evident on the anti-diagonal of the plot. We have added a brief comment on this to the manuscript.

Second, we are concerned that a plot of $g_2(t_1=t_2, t_1)$ could be easily mistaken for the more commonly plotted $g_2(\tau)$.

All in all, we believe that the additional information outweighs the possibility to compare experiment and theory more precisely.

26. Why is $g_2 < 1$ for $t_1=t_2 < 1\mu\text{s}$? In the main text it says that “the experimental signal is dominated by noise” during the rise-time. However, the simulations also show $g_2 < 1$ for $t_1=t_2 < 1\mu\text{s}$.

Reply: We agree that our wording in the article was somewhat imprecise here. In the experimental data, the onset of the pulse is dominated by noise. However, the early antibunching is caused by the onset of the system dynamics. We have rephrased the text accordingly.

27. In the second column of page 3, I don't understand the second reason for the disappearance of the correlations towards the end of the pulses, “in addition absorbers

are more likely to have undergone random Raman decay...”, can you explain it further to me please? Does it cause a “washing-out” of any correlations?

Reply: The Raman decay process, and consequently whether a superatom is saturated or not at a specific point of time, is subject to Poisson statistics. As the time of the decay is random, the superatom no longer acts as a deterministic absorber once it has decayed, but rather like a beam splitter for the coherent probe pulse with a transmission equivalent to the probability of the superatom to be excited. This results in a dominance of attenuation of the probe pulse over photon subtraction such that the photon statistics of the transmitted light are no longer affected by the interaction with the absorber.

28. In the second column of page 3, regarding “the total number of ions saturates at the sum of the three absorbers”, I understand this to mean that in Fig 3a the sum of the orange, green and red curves is the same as the blue curve. Why is this written here? I understand that depending on the time the ion reaches the MCP, the ion is interpreted as having come from either the first, second or third absorber. The detection windows have widths of 75ns (Methods). Are there gaps between the detection windows? If there are no gaps between them, then it seems obvious that the sum of the three curves will equal the blue curve. Furthermore, what is the motivation behind showing the blue curve in Fig 3a? I think that by removing the blue curve, the y-range will be reduced, and it will be easier to see how flat $\langle n_{\text{ion}} \rangle$ is for incoming photons > 10 .

Reply: The main motivation of showing the combined ion signal is indeed to highlight that the extraction of single photons by three individual superatoms adds up to the subtraction of three photons and there is a gap of 40 ns between the detection windows of the second and third absorber, but not between the first and the second as the TOF scales non-linearly with the distance from the detector (we added this information in the Methods).

The blue line indeed shows the sum of the three individual windows. While we agree that this does not contain much more additional information, we believe that the core message of our work is the ability to remove a specific number of photons from a pulse and that it is better conveyed by showing the blue curve in addition to the others, i.e. with respect for the analysis of the Mandel-Q shown in Fig. 3b where the results for the combined system are not necessarily as obvious for the reader.

29. In the second column of page 3, in the formula for the Mandel-Q parameter, I think adding parentheses would aid readability $\text{Var}(n_{\text{ion}})$...

Reply: We have added parentheses as suggested.

30. In the second column of page 3, after the formula for the Mandel-Q parameter, I think adding “which gives $-\eta$ for perfect blockade (imperfect detection leads to a binomial distribution with success probability η)” will make it easier for the reader to understand where $Q=-\eta$ comes from.

Reply: We agree that this additional explanation is helpful for readers not immediately familiar with the analysis of counting statistics and added the statement in the parentheses as suggested.

31. In the caption of Fig 3, I think the word “scaled” is more appropriate than “normalised”.

Reply: We agree and have changed the wording as suggested.

32. First column of page 4, I don’t think the phrasing “We account for these in the theoretical analysis...” is appropriate, I think it’s better to say write “We account for this in the model by increasing the Rydberg populations by a small photon-number dependent probability...”. The reason is that the curves in Fig 3c do not show the theoretical analysis, rather they show the results given by the model. The theoretical analysis involves the reasoning that there can be possible additional Rydberg excitations, etc.

Reply: We agree that this sentence should have been worded more clearly and changed it to “To account for these in the model results, we increase the excitation probabilities obtained from the Rydberg populations in the three-level model by a small, photon-number dependent probability $p_2 \langle n_{\text{ion}} \rangle$, which is independent of the superatoms’ states.” This is slightly different from the original suggestion to emphasise that the three-level model is not modified itself, but rather that we add an additional excitation probability to its results.

33. First column of page 4, the deviation of $Q/\langle n_{\text{ion}} \rangle$ from -1 is caused by double Rydberg excitation and dark counts. The double Rydberg excitation seems to be visible in Fig 3a, I think it’s easiest for the reader to digest if the double Rydberg excitation is already introduced earlier when the results of Fig 3a are discussed.

Reply: It is unfortunately unclear to us in which way double excitations should be visible in Fig. 3a. The difference in the number of detected ions for the individual superatoms is a result of the position dependence of η , which is mentioned in the text and Methods. The deviation from a flat line for higher incoming photon numbers is the result of the coherent superatom dynamics (i.e., Rabi-oscillations, see also ref. 15, i.e. Fig. 2). The theory results, which do not take the double excitation into account (unlike in panel c), also reflect these dynamics.

34. First column of page 4, “The results based on the modified values for $\langle n_{\text{ion}} \rangle$ and $\text{Var } n_{\text{ion}}$ are shown...”, at first, I read this as if the experimentally-measured values $\langle n_{\text{ion}} \rangle$ and $\text{Var } n_{\text{ion}}$ had been modified before being plotted in Fig 3c, whereas in actuality the model was modified to include effects of double Rydberg excitations and dark counts. I think changing the text to something along the lines of “The model curves shown in Fig 3c account for double Rydberg excitations and dark counts” would help.

Reply: We agree that the original wording was not very clear. We have changed it to “The model results for $\langle n_{\text{ion}} \rangle$ and $\text{Var}(n_{\text{ion}})$ shown in Fig. 3c account for double Rydberg excitations and dark counts and...”

35. First column, page 4, why does p_2 vary so much between the absorbers?

Reply: First, we attribute the variation in p_2 to position dependent variations in the control and probe parameters (answer 18). The higher value of p_2 in the centre is consistent with a higher value of Ω_c and thus excitation linewidth, which results in a lower blockade radius and an enhancement of facilitated excitation.

Second, based on geometrical considerations, ions from residual atoms located between the superatom ensembles (which are not necessarily blockaded) are also more likely to be detected as additional counts in the center time window.

36. Second column, page 4, “Our analysis underpins the hypothesis”, I don’t think the word “underpins” is appropriate. The model uses the hypothesis, the model is compared with the experimental results, and the favourable comparison supports the hypothesis.

Reply: We agree and have changed the wording to “supports” as suggested.

37. At the start of the section “Parameter optimisation and scalability”, the text reads “we determine the optimal parameters for the superatom photon absorber”. As far as I understand, the free parameters in the simplified three-level model are κR_{in} , τ and γ_D . A true optimisation would involve a three-dimensional scan of all these parameters, however this is not done here. The text should be changed to reflect this. I expect that if one was to find optimal parameters using this model, one would obtain $\kappa R_{in} \rightarrow \infty$.

I think this section clearly illustrates what are the main competing factors are towards achieving high-efficiency subtraction process.

Reply: We changed the formulation to better reflect the analysis of this section. We note though that even in the absence of Raman decay the limit $\kappa R_{in} \rightarrow \infty$ would not necessarily yield the best performance as the system enters a regime where the Rabi-oscillations of the superatom become overdamped (see ref. 15, Fig. 3).

38. Second column, page 4, “the Raman decay introduces a probabilistic component into the otherwise deterministic scheme”. Doesn’t the dephasing γ_D also introduce a probabilistic component to the scheme?

Reply: We use the word “deterministic” to underline the (almost) guaranteed absorption of a photon and “deterministic” is not a statement about the dynamics of any subsystem in our model. Since the primary purpose of our work is to shelve an excitation in the dark state, a sound definition of “deterministic” and “probabilistic” in our manuscript should consider this end goal and should not provide a synonym of unitary and dissipative dynamics. To make this distinction clearer, we changed the cited sub-sentence into “the Raman decay introduces an uncertainty about the number of absorbed photons”.

39. Second column, page 4, “This regime is bounded by three processes with independent time scales, which we indicate by dashed lines.” I think it’s worth mentioning that when the photon rate/ κ are increased too far, then second Rydberg excitations will not be blockaded, and the model breaks down.

Reply: We thank the referee for pointing out this additional limit and added a sentence with this information at the end of the paragraph “This analysis is valid until we reach large $\sqrt{\kappa R_{in}} \tau$ where we expect our model to fail, since then the blockade mechanism starts to break down and the superatom can absorb more than one photon.”

40. Fig 4a and 4c the legend text should be changed to / μs to match the rest of the text. In Fig 4b the colorbar axis is missing a label, the y-label “Photons” suggests a number, while it is a rate, and should be changed to “ R_{in} ”. The legend in Fig 4c is missing information about the different coloured curves. I think that rescaling the y-axis in Fig 4c from Dark State Population -> Dark State Population / n_{sub} would help. Right now, it is very difficult to compare the curves.

Reply: We altered the figure legends and labels as suggested. However, we tested collapsing the curves in Figure 4c to “Dark State Population / n_{sub} ” (see below; with and without dephasing curves), but we find that the resulting figure becomes too clustered and difficult to read. As an alternative we added dotted horizontal lines to Figure 4c, which makes it easier to see how close each setup is to full absorption.

41. I understand the number of subtracted photons to be + $\langle N_{\text{Raman}} \rangle$, however the caption of Fig 4 suggests that the number of subtracted photons equals . Right now, the caption text does not explain Fig 4 well. To understand this figure, the reader needs to read the main text. The caption text should be developed.

Reply: We improved the figure caption for Figure 4a to remove this source of confusion. Additionally, we added a remark that Figure 4b corresponds to the photon absorption probability. We agree that the figure caption leaves some aspects of the figure unanswered; yet, this

cannot be circumvented without bloating the caption. We compromised, by giving only the most central information as well as the numerical constants.

42. First column of page 5, I suggest changing the text from the idealistic “in the absence of Raman decay, the absorption probability can be made arbitrarily large by increasing τ ” to something more physical “by decreasing the Raman decay rate, higher absorption probabilities can be achieved.”

Reply: We thank the referee for the alternative formulation and included it in the manuscript.

43. Second column, page 5, “An immediate application is number-resolved detection of up to n_{sub} photons based on the number of absorbers in a Rydberg state ... by increasing n_{sub} well beyond the expected photon number, a weak photon-absorber coupling κ could be compensated”. It would be nice if there was data to support the “immediate” applicability of the method. However, Fig 4b doesn’t support this, given that the pulse considered involved 20 photons, which is greater than the number of absorbers.

I think a reader would like to know how many absorbers would be needed to resolve the number of photons in a pulse of up to ~ 5 photons with a fidelity of $\sim 90\%$ (with reasonable experimental parameters)? I would like to see a 2D colour graph with n_{sub} on the x-axis, the number of photons on the y-axis, and the colour indicating the expected fidelity (or infidelity) of a photon-number-resolving measurement. I expect that a photon-number-resolving measurement device using an array of Rydberg super-atoms would have difficulty out-performing a bunch of beam splitters and single-photon detectors.

Reply: We agree that a more in depth discussion including the proposed plot would be very interesting, but consider this to be beyond the scope of this work and more appropriate for a follow-up paper that focuses on number-resolved photon detection. To avoid the impression that number-resolved photon-detection can be achieved without any further work at all, we rephrased the statement as follows: “A more readily implementable application is number-resolved detection of up to n_{sub} photons...”

A scheme based on beamsplitters and single-photon counters would be limited by the quantum efficiency and deadtime of the detectors (which is of course typically larger than η in this work, but this could be overcome, see next answer). An important difference is however that in a chain of superatoms a photon that is not absorbed by the first can still be absorbed by a subsequent superatom, while a photon that is incident on a photon detector, but not detected is lost and will not be counted.

44. Second column, page 5, the detection efficiency “could be significantly improved by replacing the MCP ... or using optical detection”. How far is it reasonable to expect that the detection efficiency can be improved?

Reply: For optical detection, a optical transistor scheme as demonstrated in ref.s 25 and 26 could be used. In these, a single Rydberg excitation can suppress the transmission of several tens of photons (see Nat. Commun. 7, 12480 (2016) for the state-of-the-art) such that close to unity detection efficiency is realistic. For MCPs, detection efficiencies of order 80-90% have been reported (Rev. Sci. Instr. 89, 045112 (2018)).

45. Second column, page 5, “Raman decay should still be strongly suppressed”, it isn’t clear when reading this whether “should” is meant in a passive or active sense. Something along the lines of “A high detection efficiency also requires suppressing Raman decay, which acts to reduce the efficiency.” Might be better.

Reply: We have rephrased the statement to avoid ambiguity: “Meanwhile, it is still important to minimise Raman decay as it reduces the detection efficiency for each absorber.”

46. First column, page 6, “in principle deterministic scheme”, as I mentioned earlier, I think both Raman decay and the engineered dephasing make the scheme non-deterministic.

Reply: We removed this phrase and simply state that the Raman decay gives probabilistic fluctuations. Please see also reply 38 regarding the definition of deterministic as intended by us.

47. Methods, second column page 6, do you know whether the width of the time window due to jitter of the MCP or the distribution of the ion flight times?

Reply: The width of the time window is determined by the distribution of ion flight times. In a single shot this is determined by the initial position and velocity distribution of the atoms in an ensemble. For technical reasons related to the fast high-voltage switching of the ionisation field, there is also a drift in the ionisation voltage applied in each of the 500 consecutive experiments before new ensembles are prepared, which leads to a variation in the arrival time for each of these shots that exceeds the width of the distribution obtained in a single shot. In our data evaluation, we choose the time windows for each dimple wide enough such that it can stay the same for all 500 repetitions to minimise the need for post-processing.

Referee 2

The authors report a study of sequential single-photon subtraction by Rydberg superatoms. The central effect of the study - single-photon subtraction - is expected to increase the value of g_2 at small delays, and the authors measure such an effect in a system of three superatoms. The experimental results are convincing and supported by extensive theoretical analysis. The technique they demonstrate can be further refined and applied for more involved manipulations of non-classical states of light and atoms. I recommend the manuscript for publication in its present form.

Reply: Thank you for your careful review, the positive and encouraging feedback, and the recommendation to publish the article as-is.

Referee 3

The authors report an experimental and theoretical study of controlled multi-photon subtraction with cascaded Rydberg superatoms.

Their scheme exploits the Rydberg blockade in small atomic ensembles, which means that merely one atom out of many can be excited to a Rydberg state. This leads to a delocalised Rydberg excitation, which dephases quickly. This mechanism strongly suppresses the reemission of the absorbed photon into the mode of the excitation beam. This realises a single photon absorber, and the authors investigate and characterise a system that is composed of three such concatenated absorbers.

The work is elegant, timely and interesting - especially when considering that such passive devices may find applications in optical quantum information processing and communication. The theory strongly supports the experimental findings and a serious account of possible error sources and performance limitations is given. The associated discussion is not only relevant for the particular system of Rydberg superatoms, but highlights general issues when building quantum devices that are based on few-level systems subject to decoherence.

I do not see any shortcoming of this work and - due to the timeliness, relevance and high quality of the results - I recommend publication in Nature Communications.

Reply: We thank you for your careful review, the positive feedback, and the recommendation to publish the article in Nature Communications.

1. The dashed lines in the inset of Fig 2a are too faint.

2. I think it would be good to rethink the presentation of the data in panel 2b. It is very hard to discriminate the two colored curves that are overlaid in each sub-panel.

Reply (to both points): We agree that these aspects of the presentation could be improved and have modified the figure.

3. It might be worth pointing out that the excitation of a Rydberg superatom together with ionisation realises a single ion source, e.g. discussed in [Physical Review Letters 110, 213003 (2013)].

Reply: This is indeed worth pointing out and we have added a brief statement the suggested reference (alongside reciting ref. 37)

Reviewers' Comments:

Reviewer #1:

Remarks to the Author:

I'm content with your reply.

I have some further comments:

In response to comment 2 you removed text from the abstract which said that the simulations indicate that the system will scale well with n_{sub} . I think the first sentence of the second paragraph of column 1 on page 5 should be similarly changed.

I think the caption of Fig 3 now has a typo, it should read "scaled" instead of "scale".

Regarding comment 39, now the manuscript reads "large R_{in}/κ ". Is this meant to be read "large R_{in} divided by κ " (which I think is incorrect) or "large R_{in} or large κ "? I think the text you wrote in your rebuttal ($\sqrt{\kappa R_{\text{in}}}$) is better.

Yours sincerely,

Gerard Higgins

Referee 1

I'm content with your reply.

I have some further comments:

In response to comment 2 you removed text from the abstract which said that the simulations indicate that the system will scale well with n_{sub} . I think the first sentence of the second paragraph of column 1 on page 5 should be similarly changed.

Reply: Thank you for the comment. We have removed the statement as suggested.

I think the caption of Fig 3 now has a typo, it should read "scaled" instead of "scale".

Reply: Thank you for pointing out this typo. It has been corrected.

Regarding comment 39, now the manuscript reads "large R_{in}/κ ". Is this meant to be read "large R_{in} divided by κ " (which I think is incorrect) or "large R_{in} or large κ "? I think the text you wrote in your rebuttal ($\sqrt{\kappa R_{\text{in}}}$) is better.

Reply: Thank you for noting the ambiguity of the statement. Of course, we mean "large R_{in} or large κ ". We have modified statement to match the rebuttal as suggested.

Yours sincerely,
Gerard Higgins

Reply: Dear Gerard, once more thank you very much for your extraordinarily detailed review and feedback.